# Reconstructing Elemental Carbon Long-Term Trend in the Po Valley (Italy) from Fog Water Samples

**Stefania Gilardoni [1], Leone Tarozzi [2], Silvia Sandrini [3], Pierina Ielpo [4], Daniele Contini [4], Jean-Philippe Putaud [5], Fabrizia Cavalli [5], Vanes Poluzzi [6], Dimitri Bacco [6], Cristina Leonardi [7], Alessandra Genga [8], Leonardo Langone [1] and Sandro Fuzzi [3,*]**

1 Institute of Polar Sciences, National Research Council, 40129 Bologna, Italy; stefania.gilardoni@cnr.it (S.G.); leonardo.langone@cnr.it (L.L.)
2 Institute of Marine Sciences, National Research Council, 40129 Bologna, Italy; leone.tarozzi@cnr.it
3 Institute of Atmospheric Sciences and Climate, National Research Council, 40129 Bologna, Italy; s.sandrini@isac.cnr.it
4 Institute of Atmospheric Sciences and Climate, National Research Council, 73100 Lecce, Italy; p.ielpo@isac.cnr.it (P.I.); d.contini@isac.cnr.it (D.C.)
5 European Commission, Joint Research Centre, 21027 Ispra, Italy; jean.putaud@ec.europa.eu (J.-P.P.); fabrizia.cavalli@ec.europa.eu (F.C.)
6 Regional Agency for Prevention, Environment and Energy of Emilia-Romagna, 40122 Bologna, Italy; vpoluzzi@arpa.emr.it (V.P.); dbacco@arpae.it (D.B.)
7 Air Pollution Institute, National Research Council at the Ministry of Environment, Land, and Sea, 00147 Rome, Italy; cristina.leonardi@cnr.it
8 Department of Biological and Environmental Sciences and Technologies, University of Salento, 73100 Lecce, Italy; alessandra.genga@unisalento.it
* Correspondence: s.fuzzi@isac.cnr.it

**Abstract:** Elemental carbon (EC), a ubiquitous component of fine atmospheric aerosol derived from incomplete combustion, is an important player for both climate change and air quality deterioration. Several policy measures have been implemented over the last decades to reduce EC emissions from anthropogenic sources, but still, long-term EC measurements to verify the efficacy of such measurements are limited. In this study, we analyze the concentration of EC suspended in fog water samples, collected over the period 1997–2016 in a rural background site of the southern Po Valley. The comparison between EC in fog water and EC atmospheric aerosol concentration measured since 2012 allowed us to reconstruct EC atmospheric concentration from fog water chemical composition dating back to 1997. The results agree with the EC atmospheric observations performed at the European Monitoring and Evaluation Program (EMEP) station of Ispra in the northern part of the Po Valley since 2002, and confirm that the Po Valley is a pollution hotspot, not only in urban areas, but also in rural locations. The reconstructed trend over the period 1997–2016 indicates that EC concentration during the winter season has decreased on average by 4% per year, in agreement with the emission reduction rate, confirming the effectiveness of air quality measures implemented during the past 20 years.

**Keywords:** elemental carbon; atmospheric aerosol; fog; Po Valley

## 1. Introduction

The occurrence in the atmosphere of elemental carbon (EC), referred to also as black carbon (BC), is relevant from both climatic and human health standpoints [1,2]. The terms elemental carbon and black carbon are used interchangeably, and refer to the strongly light-absorbing atmospheric

aerosol component emitted during incomplete combustion. EC and BC differ for the methodologies used for their quantification, i.e., thermal refractivity and light absorption, respectively. The Special Report of the Intergovernmental Panel on Climate Change (IPCC) released in 2018 [3] identifies emission reduction of black carbon, together with methane, an unavoidable measure to keep the global average temperature increase within 1.5 °C by the end of the century [3]. The relevance of BC as a short-lived climate forcer and an atmospheric pollutant was already highlighted by the Gothenburg Protocol amendments, signed in 2012 and entered into force in 2019. The amendments introduced particulate matter (PM) and BC into the list of air pollutants whose emission reductions would lead to co-benefits for both climate and human health. This amendment also supports the European Directive on National Emission Ceilings (2016/2284/EU), which sets national emission reduction targets for PM and, indirectly, BC.

Design and implementation of effective policy measures aiming at reducing EC/BC ambient concentration require long-term monitoring to verify the efficiency of the implemented measures.

In the United States, the Chemical Speciation Network (CSN) and the Interagency Monitoring of Protected Visual Environments (IMPROVE) network record EC from 2000, and total carbon (TC) from 1989, in over 300 sites [4]. These records show that TC has decreased over the period 1990–2010, with the exception of the western states in summer, likely due to the effects of more frequent wildfire episodes. In Europe, the European Monitoring and Evaluation Program (EMEP) network started introducing EC measurements in the so-called "level-2" sites in 2004, and BC in 2010 [5].

An alternative approach to reconstruct long-term trends of EC in the atmosphere is to investigate natural archives of atmospheric composition data, such as sediments and ice cores. Housain et al. [6] reconstructed EC atmospheric concentration back to 1835 at Whiteface Mountain (NY, USA), using both EC lake sediment contents and recent atmospheric EC measurements. They observed that present-day EC concentration is about two times higher than pre-industrial levels, with the highest concentrations over the period 1917–1930. The comparison with reconstructed emissions indicate that, generally, the atmospheric EC follows changes in fuel consumption, with the most significant decreases after 1930, due to the economic crisis, and in the late 1970s, due to the progressive replacement of coal with oil, and the introduction of regulations on transport emissions [6]. Legrand et al. [7] measured the EC concentration trend in ice core collected at Col du Dôme, in the French Alps. They observed that EC concentration increased sharply after the 1940s, and started decreasing after the 1970s. This trend was then employed to validate atmospheric emission inventory data back to the 1920s [8].

At the European level, the Ambient Air Quality Directive 2008/50/CE does not set limits for EC ambient concentration, nevertheless, it foresees the chemical speciation of $PM_{2.5}$ in rural background locations, including the determination of EC. Therefore, several European countries started introducing EC in the national and regional air quality monitoring networks [9] in the late 2000s. This will provide the opportunity in the near future to investigate EC trends across different environments, but, at the moment, long-term data records are still very limited.

In this study, we aim to reconstruct the atmospheric EC concentration trend in the Po Valley, Italy, based on chemical analysis of fog water composition. At the research station of San Pietro Capofiume (SPC; 44.66° N, 11.62° E), in the southern part of the Po Valley, fog water sampling has been routinely performed since the late 1990s. The fog season in the Po Valley spans from November through to March. Aerosol EC is scavenged within the fog droplets and can be measured, after fog water sample filtration, by thermal or thermal-optical analysis. In this way, we have obtained a long-term data record of aerosol EC concentration at this site, although limited to the winter season, when fog is present. Over a period of about 20 years, this would nonetheless allow a trend reconstruction of EC atmospheric concentration.

## 2. Experiments

### 2.1. Sampling Sites

Measurements discussed in this paper were performed at two rural sites in the Po Valley, San Pietro Capofiume (SPC) and Ispra (Figure 1a). The Po Valley, Northern Italy, is a highly polluted region [10], due to significant anthropogenic emissions from both the agricultural and industrial sectors and from transportation. In addition, one fourth of the national population lives in this area, with a population density almost double the national average. Finally, the valley is surrounded to the north, west, and south by mountain ranges that prevent pollution dispersion and favor accumulation, especially during the cold season.

SPC is a rural background site, part of the air quality monitoring network of the Regional Environmental Protection Agency (ARPAE). The site is located about 30 km northeast of Bologna, and is surrounded by agricultural fields and small villages. It is characterized by stable meteorological conditions with high relative humidity, low wind speed (generally below 2 m s$^{-1}$), and low temperature during the winter season, leading to frequent fog events [11]. Sampling site and air quality characterization during winter at SPC have been described previously [12,13].

Ispra is a regional background site located in the northwestern part of the Po Valley, about 60 km north of Milan [14]. The nearest urban areas are Novara, 40 km to the south, and Varese, 20 km to the west. The site has been part of the European Monitoring and Evaluation Program (EMEP) since 1985. The site is affected by polluted air masses from the Po Valley in southerly wind conditions, while clean air masses are observed during northerly wind periods [15].

The extensive compilation of PM$_{2.5}$ concentrations and carbonaceous aerosol measurements across different sites in Italy show that the two sites are representative of the rural Po Valley [16].

### 2.2. Fog Water Collection

Fog water samples were collected at the rural site of SPC. Giulianelli et al. [13] investigated fog occurrence and fog water chemical composition trends during the last twenty years at SPC. They reported a decreasing trend in fog occurrence frequency, as well as a reduction in the amount of sulfate, nitrate, and ammonium dissolved in fog water. Routine fog water collection started in winter 1997, and is performed systematically from November to March.

Fog water is collected with an automated home-made collector, described in detail by Fuzzi et al. [17]. During each fog event, a single fog water sample is collected. The collector is equipped with a fan located on the edge of a tunnel, that creates a flow of about 17 m$^3$ min$^{-1}$. The air stream generated by the fan passes through a grid made of parallel stainless-steel strings, where fog droplets coalesce, to eventually fall into a funnel and then into the sampling bottle. The 50% collection efficiency of each string is about 3 μm.

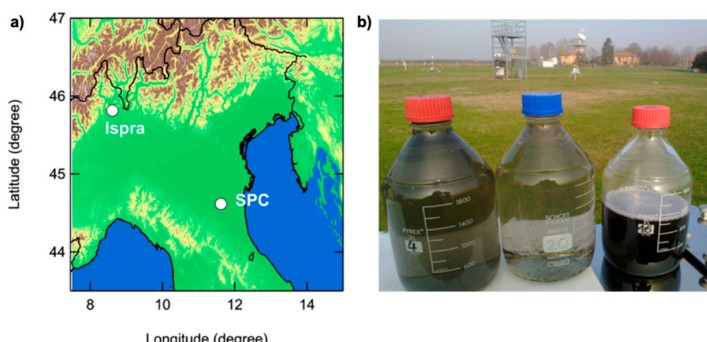

**Figure 1.** Map of Northern Italy, reporting the measurement sites San Pietro Capofiume (SPC) and Ispra (**a**); example of fog water samples collected at SPC (**b**). The different content of elemental carbon (EC) is visually shown by the blackness of the different samples.

The fog water collector is driven by a computer program specifically designed to activate the fog sampling when the temperature is higher than 1 °C (to avoid freezing conditions), and liquid water content (LWC) is larger than 0.08 g m$^{-3}$ (set at arbitrary threshold, and corresponding to a dense fog condition). LWC is continuously measured with a particle volume monitor (PVM-100) at 1-min time resolution.

The data here reported refer to the winter seasons from 1997 to 2016. The number of fog samples collected during each season varied between 7 and 32, with an average of 20 samples per season (428 samples over 20 years).

### 2.3. Quantification of Carbonaceous Particles Suspended in Fog Water Samples

Immediately after sampling, 10 mL of fog water samples are filtered through a quartz fiber filter (PALL QAO-UP 2500 filters, Ø 47 mm, Port Washington, NY, USA) with a glass vacuum filtration device. Filters are then stored at −4 °C prior to analysis.

Samples collected between 1997 and 2001 were analyzed for total carbon (TC) and EC by double thermal analysis. The analysis of filter punches exposed for several minutes to a stream of pure O$_2$ at 340 °C led to the determination of EC, and the analysis of filter punches without thermal treatment led to the determination of TC. The temperature of 340 °C was chosen to remove organic aerosol without oxidation of EC [18], and to avoid the potential effects of organic aerosol charring and EC overestimation. No direct comparison of this double thermal analysis procedure with more recently standardized protocols is available for insoluble carbonaceous material.

From 2002 to 2011, we measured TC concentrations in filtered fog water samples using an EA Flash2000 Thermo Fisher Scientific. For each sample, two to four replicate analyses of 6 or 9 mm diameter punches were performed. At the beginning of each sequence of analysis, three capsules of Atropine (IAEA standard) and one of USGS-40 were analyzed. Then, for every 8–10 samples, one capsule of an Adriatic sediment used as laboratory reference was analyzed in order to compensate for potential instrument drift, and as a quality control measure. The average standard deviation of each measurement, determined by replicate analyses of the same sample, was ±0.07% for TC.

From 2012, the EC and OC concentrations collected on the filters were quantified by thermal-optical analysis (Sunset, Laboratory Inc., USA), using the NIOSH protocol [19].

### 2.4. Analysis of EC in Atmospheric Aerosol Samples

PM$_{2.5}$ aerosol samples for EC quantification have been collected routinely at SPC, starting in winter 2012, with 24-h aerosol samples collected from midnight. Sampling is performed on quartz fiber filters (PALL QAO-UP 2500 filters, Ø 47 mm), with a low-volume sampler (TCR Tecora) at a flow-rate of 16.7 L min$^{-1}$ [20]. Filters are pre-baked at 800 °C for 5 h in order to remove any carbonaceous material. EC is then quantified by thermal-optical analysis in transmittance (TOT) using the EUSAAR2 protocol [21,22].

## 3. Results and Discussion

### 3.1. TC and EC Seasonal Trends

Figure 2 reports the inter-annual variability of insoluble particulate TC measured in fog water over the whole sampling period. Each box corresponds to a fog season starting in November and ending in March of the following year. The year identifying the fog season in the graph corresponds to the beginning of the fog season. Season-averaged TC values ranged from 2.9 μg mL$^{-1}$ (24 samples) to 16.6 μg mL$^{-1}$ (7 samples), while single fog event TC concentrations ranged between 0.4 μg mL$^{-1}$ and 47.0 μg mL$^{-1}$. In order to test if a significant time trend could be detected over the investigated period, we calculated the Theil–Sen estimator of the seasonal TC concentration. The resulting slope was −0.08 μg mL$^{-1}$ year$^{-1}$ with a median absolute deviation (MAD) equal to 0.23 μg mL$^{-1}$ year$^{-1}$. The large MAD compared to the slope confirms that no time trend can be detected in the concentration of particulate TC suspended in fog water over the last twenty years. A previous analysis of fog water

chemical composition trends clearly showed a significant decrease in water-soluble inorganic species dissolved in fog water [13]. In particular, sulfate, nitrate, and ammonium concentration decreased by 76%, 43%, and 55%, respectively, starting from the late 1990s. On the contrary, water-soluble organic carbon did not show any significant trend [13]. The results presented here also indicate that the insoluble carbonaceous particles suspended in fog water follow the same behavior as fog water-soluble carbon.

During the first years of fog collection and analysis (from 1997 to 2000), and then more recently during the last five fog seasons analyzed (from 2012 to 2016), the characterization of carbonaceous particles suspended in fog water allowed us to quantify the EC fog water content, together with insoluble particulate TC. Over the whole measurement period, EC averaged 1.2 µg mL$^{-1}$, ranging between 0.1 and 7.1 µg mL$^{-1}$ in single event samples. The trend of EC concentrations measured in fog water shows a large year-to-year variability, with seasonal averages varying from 0.5 µg mL$^{-1}$ to 2.7 µg mL$^{-1}$. Despite the limited dataset, the variability of seasonal averages over the different years is within the single seasonal variability, and no long-term EC trend can be detected.

In order to verify the similarity of the two datasets when EC measurements are available (before 2000 and after 2012), we first tested the normal distribution of EC to TC ratios, and then performed the Welch *t*-test. Figure 3 shows the Q-Q plot of EC/TC ratios observed during the two periods, reporting on the y-axis the measured ratios, and on the x-axis the distribution of the data around the mean (quantified by z-scores or number of standard deviations). The plots indicate that in both periods, the ratios were distributed around the mean with a variability of about two times its standard deviation. Over the period 1997–2000, the distribution was skewed towards larger values, showing outliers. After removing the outliers observed during the first part of the experiment (4 out of 73 samples), which could possibly come from the different analytical method used at that time, the Welch *t*-test indicates that the two periods showed identical EC to TC ratios at a confidence interval of 99.95%.

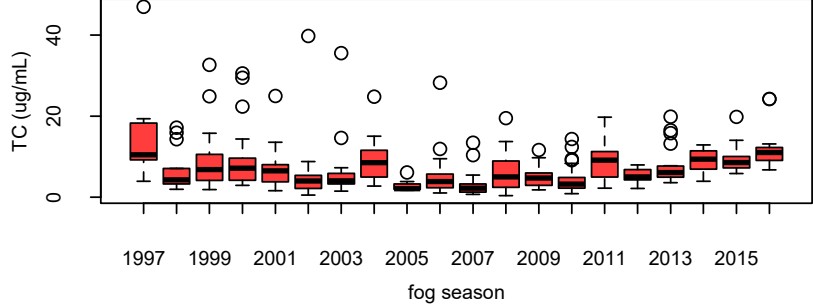

**Figure 2.** Box and whisker plots of total carbon (TC) concentration in fog water during the different fog seasons.

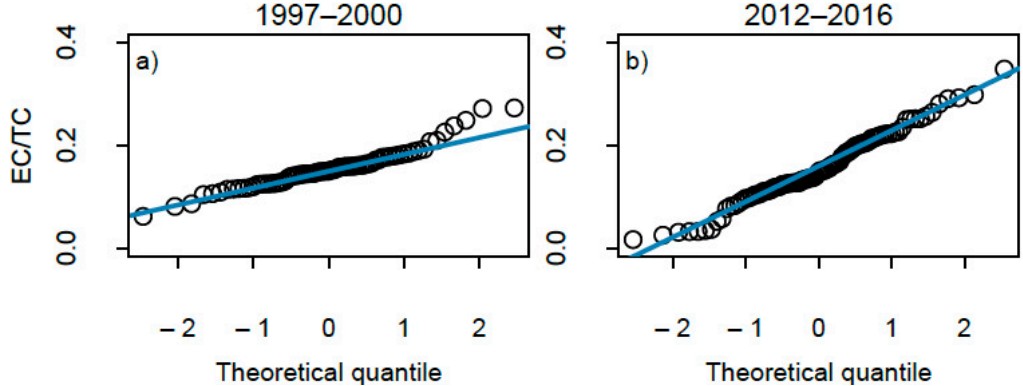

**Figure 3.** Q-Q plot of EC to total carbon ratio from 1997 to 2000 (panel **a**), and from 2012–2016 fog season (panel **b**). The blue line indicates the theoretical normal distribution.

The average EC to TC ratio calculated over the entire dataset (0.16 ± 0.01) was then used to estimate the concentration of EC suspended in fog water from the TC concentration between 2000 and 2012 when no direct EC measurements were available. The seasonal averages of measured EC are reported in Figure 4 in red, while the estimated EC seasonal averages are reported in cyan.

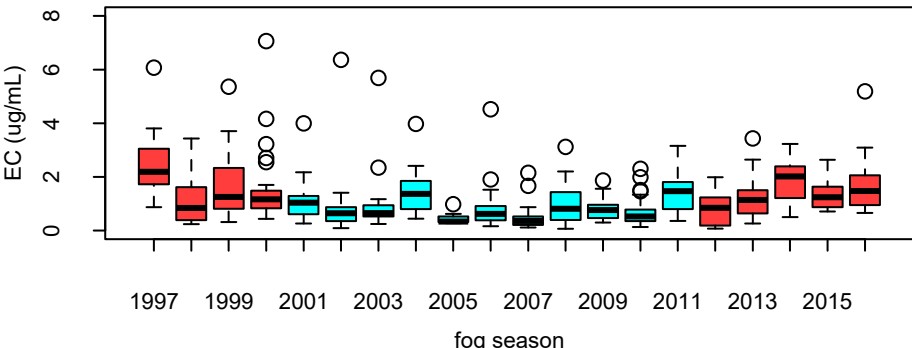

**Figure 4.** Box and whisker plots of EC concentration in fog water during the different fog seasons; measured concentrations are in red, and estimated concentration in cyan.

### 3.2. Atmospheric EC Concentrations

We calculated the atmospheric loading of EC captured by fog droplets, by multiplying the concentration of EC suspended in fog water by the atmospheric liquid water content (LWC). The obtained concentration could be considered as a proxy of EC atmospheric concentrations, under the assumption that EC was completely scavenged by fog droplets. Previous observations showed that the EC scavenging coefficient, e.g., the fraction of EC concentration that is removed from the atmosphere by fog, is actually far from 1. Gilardoni et al. [12] observed that the EC scavenging coefficient in fog can vary significantly from one event to the other, with scavenging coefficients ranging between 0.3 and 0.6. Hallberg et al. [23] reported, instead, a significantly lower fog scavenging coefficient, equal to 0.06. It is likely that, since EC is expected to be found in smaller aerosol particles (generally below 300 nm), the efficiency of scavenging depends strongly on the particles' mixing state [24], which varies over time and space.

An accurate determination of EC scavenging efficiency would require the comparison of atmospheric EC concentration right before and after fog formation [12]. In this study, we are more interested in identifying a conversion parameter to extrapolate atmospheric seasonal variability of atmospheric EC concentrations from long-term fog composition measurements. With this purpose, we calculated the average values of EC concentration in $PM_{2.5}$ aerosol samples, over each whole fog season (from November to March), and compared them with the seasonal averages of EC fog water concentration, adjusted for LWC. The ratios vary between 0.16 and 0.49, with an average of 0.28 ± 0.05. Based on this conversion factor, we derived the EC atmospheric concentration from fog water composition back to 1997.

### 3.3. Long-Term Trends of Atmospheric EC Concentrations

The fog-season averaged EC concentration estimated for SPC is shown in Figure 5a by the blue bars. The error bars in Figure 5a take into account the seasonal variability, as well as the uncertainty associated with the assumption of a constant EC to TC ratio, to calculate EC fog water concentration, and a constant conversion factor, to calculate atmospheric concentrations. The seasonal averages range between 0.8 and 4.2 $\mu g\ m^{-3}$. For comparison, Table 1 reports the average EC concentrations observed in $PM_{2.5}$ and $PM_{10}$ aerosol samples collected at different European sites. Sandrini et al. [16] reported a survey of EC measurements at different urban and rural sites in the Po Valley, mainly between 2005 and 2008. The seasonal EC average ranged between 1 and 2 $\mu g\ m^{-3}$ at the rural sites, and between 2 and 4 $\mu g\ m^{-3}$ at the urban sites ([16] and references therein). Over the same period, the seasonal EC

average estimated at SPC varied between 1 and 3.4 μg m$^{-3}$, in agreement with previously-reported EC atmospheric concentrations.

**Table 1.** Mean EC concentrations observed in PM$_{2.5}$ and PM$_{10}$ fraction in different European sites (re: regional background, r: rural, s: suburban, ur: urban).

| Site | Country | Observation Period | Size Fraction | EC (μg m$^{-3}$) | Reference |
|---|---|---|---|---|---|
| Oasi le Bine (r) | Italy | 2007–2008 | PM$_{2.5}$ | 0.9 | [16] |
| Genova (ur) | Italy | 2010–2011 | PM$_{2.5}$ | 1.2 | [16] |
| Bari (ur) | Italy | Spring 2007 | PM$_{2.5}$ | 1.7 | [16] |
| Milano (ur) | Italy | 2005–2007 | PM$_{2.5}$ | 2.3 | [16] |
| Mantova (ur) | Italy | 2005–2007 | PM$_{2.5}$ | 1.3 | [16] |
| Brescia (ur) | Italy | 2005–2007 | PM$_{2.5}$ | 1.7 | [16] |
| Birkenes (re) | Norway | Fall 2008 Winter/Spring 2009 | PM$_{2.5}$ | 0.1 0.10 | [25] |
| Ispra (re) | Italy | Fall 2008 Winter/Spring 2009 | PM$_{2.5}$ | 1.5 1.5 | [25] |
| K-puszta (re) | Hungary | Fall 2008 Winter/Spring 2009 | PM$_{2.5}$ | 1.2 0.77 | [25] |
| Kosetice (re) | Czech Rep. | Fall 2008 Winter/Spring 2009 | PM$_{2.5}$ | 0.49 0.32 | [25] |
| Lille Valby (r) | Denmark | Fall 2008 Winter/Spring 2009 | PM$_{2.5}$ | 0.46 0.37 | [25] |
| Mace Head (re) | Ireland | Fall 2008 Winter/Spring 2009 | PM$_{2.5}$ | 0.12 0.11 | [25] |
| Melpitz (re) | Germany | Fall 2008 Winter/Spring 2009 | PM$_{2.5}$ | 0.54 0.40 | [25] |
| Montelibretti (r) | Italy | Fall 2008 Winter/Spring 2009 | PM$_{2.5}$ | 0.97 1.0 | [25] |
| Payerne (r) | Switzerland | Fall 2008 Winter/Spring 2009 | PM$_{2.5}$ | 0.59 0.66 | [25] |
| Birkenes (re) | Norway | 2008–2011 | PM$_{2.5}$ PM$_{10}$ | 0.1 0.1 | [26] |
| Melpitz (re) | Germany | 2010–2011 | PM$_{2.5}$ PM$_{10}$ | 0.6 0.8 | [26] |
| Montseny (re) | Spain | 2008–2011 | PM$_{2.5}$ PM$_{10}$ | 0.2 0.3 | [26] |
| Ispra (re) | Italy | 2008–2011 | PM$_{2.5}$ PM$_{10}$ | 1.6 2.0 | [26] |
| Kosetice (re) | Czech Rep. | 2009–2011 | PM$_{2.5}$ | 0.6 | [26] |
| Aspvreten (re) | Sweden | Winter seasons 2010–2011 | PM$_{10}$ | 0.20 | [27] |
| Birkenes (re) | Norway | Winter seasons 2010–2011 | PM$_{10}$ | 0.082 | [27] |
| Finokalia (re) | Greece | Winter seasons 2008–2010 | PM$_{10}$ | 0.18 | [27] |
| Harwell (re) | United Kingdom | Winter 2010 | PM$_{10}$ | 0.39 | [27] |
| Ispra (re) | Italy | Winter seasons 2008–2011 | PM$_{2.5}$ | 2.03 | [27] |
| Melpitz (re) | Germany | Winter seasons 2008–2010 | PM$_{10}$ | 0.55 | [27] |
| Montseny (re) | Spain | Winter seasons 2008–2011 | PM$_{10}$ | 0.23 | [27] |
| Puy de Dome (re) | France | Winter seasons 2008–2010 | PM$_{10}$ | 0.067 | [27] |
| Vavhill (re) | Sweden | Winter seasons 2010–2011 | PM$_{10}$ | 0.30 | [27] |
| Kosetice (re) | Czech Rep. | Winter seasons 2013–2016 | PM$_{2.5}$ | 0.83 | [28] |
| Krynica Zdroj | Poland | Winter 2017 | PM$_{10}$ | 1.34 | [29] |
| Zloty Potok (r) | Poland | Winter 2013 | PM$_{2.5}$ | 2.17 | [30] |
| Racibórz (s) | Poland | Winter seasons 2011–2012 | PM$_{2.5}$ | 3.59 | [30] |
| Puszcza Boreka (re) | Poland | Winter 2011 | PM$_{2.5}$ | 0.84 | [30] |
| Zielonka (r) | Poland | Winter 2011 | PM$_{2.5}$ | 1.25 | [30] |
| Szczecin (r) | Poland | Winter 2013 | PM$_{2.5}$ | 1.67 | [30] |
| Trzebinia (r) | Poland | Winter 2013 | PM$_{2.5}$ | 3.97 | [30] |
| Prague (r) | Czech Rep. | Winter 2003 | PM$_{2.5}$ | 1.69 | [30] |

A multi-site analysis performed across European rural sites at different latitudes in fall 2008 and winter 2009 reports EC concentrations ranging from 0.1, in Northern Europe, to 1.5 μg m$^{-3}$, in Southern Europe [25]. Cavalli et al. [26] and Zanatta et al. [27] investigated the seasonal average values of EC for several rural background sites across Europe, from multi-year observations. The EC winter averages, excluding those reported for Ispra, ranged between 0.1 and 0.8 μg m$^{-3}$. Higher EC concentrations are typically observed in Northeastern Europe during the heating season (typically from September to March). Single season experiments report average EC values varying between 0.8 and 3.9 μg m$^{-3}$ [28–30]. Ispra is a regional background site strongly affected by anthropogenic emissions, located in the northern part of the Po Valley, as previously observed based on back trajectory statistical analysis [14,31]. The average atmospheric EC concentration measured at Ispra during the period 2008–2011 was 1.6 μg m$^{-3}$ in PM$_{2.5}$ samples and 2 μg m$^{-3}$ in PM$_{10}$ samples [26], thus higher than the other rural Western Europe sites, and comparable to concentrations in Eastern Europe sites. Similarly, during the winter seasons of 2008–2011, the EC concentration at SPC, estimated from fog water composition, was 1.4 μg m$^{-3}$. Altogether, these results confirm that the Po Valley is a pollution hotspot, not only in its highly urbanized environment, but also in rural locations.

Figure 5a reports the EC atmospheric concentration from long-term measurements performed at the EMEP site of Ispra, together with the SPC data. The long-term trend of EC concentration in SPC derived from fog water composition is compared with the atmospheric EC observations in Ispra, to show the consistency of the two series. Ispra monitoring site, like SPC, is located in the Po Valley; it is a regional background site representative of the northern part of the valley, and is characterized by one of the longest time records of atmospheric EC measurements in Europe. The comparability of Ispra data from 2002 to 2004 with the following seasons is limited, since EC was measured with a flash heating procedure, which did not allow for charring correction, and data coverage was also lower. The EC/TC ratio determined with the flash heating procedure (without charring correction) used from 2002 to 2004 was compared with the EC/TC ratio determined with a thermal-optical method implementing the EUSAAR2 thermal protocol against PM$_{2.5}$ samples in 2005, and EC data reported in Figure 5 are corrected accordingly. The seasonal EC averages estimated at SPC are comparable or slightly lower than the seasonal EC averages measured at Ispra. The higher concentrations reported in Ispra can be attributed to the strong influence of anthropogenic emissions observed at this site, relative to other rural sites in the Po Valley [31]. During the overlapping period, the highest EC concentrations at SPC in the years 2006 and 2011 coincide with the high EC concentrations at Ispra. These results support the robustness of the procedure employed here to derive atmospheric concentrations from fog water composition data.

We have analyzed the trend of atmospheric EC concentration over the 20-year period from 1997 to 2016 at SPC (Table 2). First, we calculated the Kendall's tau, which defines the strength of the rank correlation between the variables "fog season year" and "EC concentration". The tau value indicates that the null hypothesis, i.e., EC varies independently from the year, has to be rejected at the 95% confidence level. The Theil-Sen's slope quantifies the negative trend, which corresponds to a 0.06 μg m$^{-3}$ decrease in EC per year, with a relatively small MAD value (0.03). The same analysis, applied to a shorter time interval (from 2003 to 2016), identifies a larger decreasing trend at the site of Ispra, resulting from the larger values observed between 2006 and 2010 in the northern part of the Po Valley.

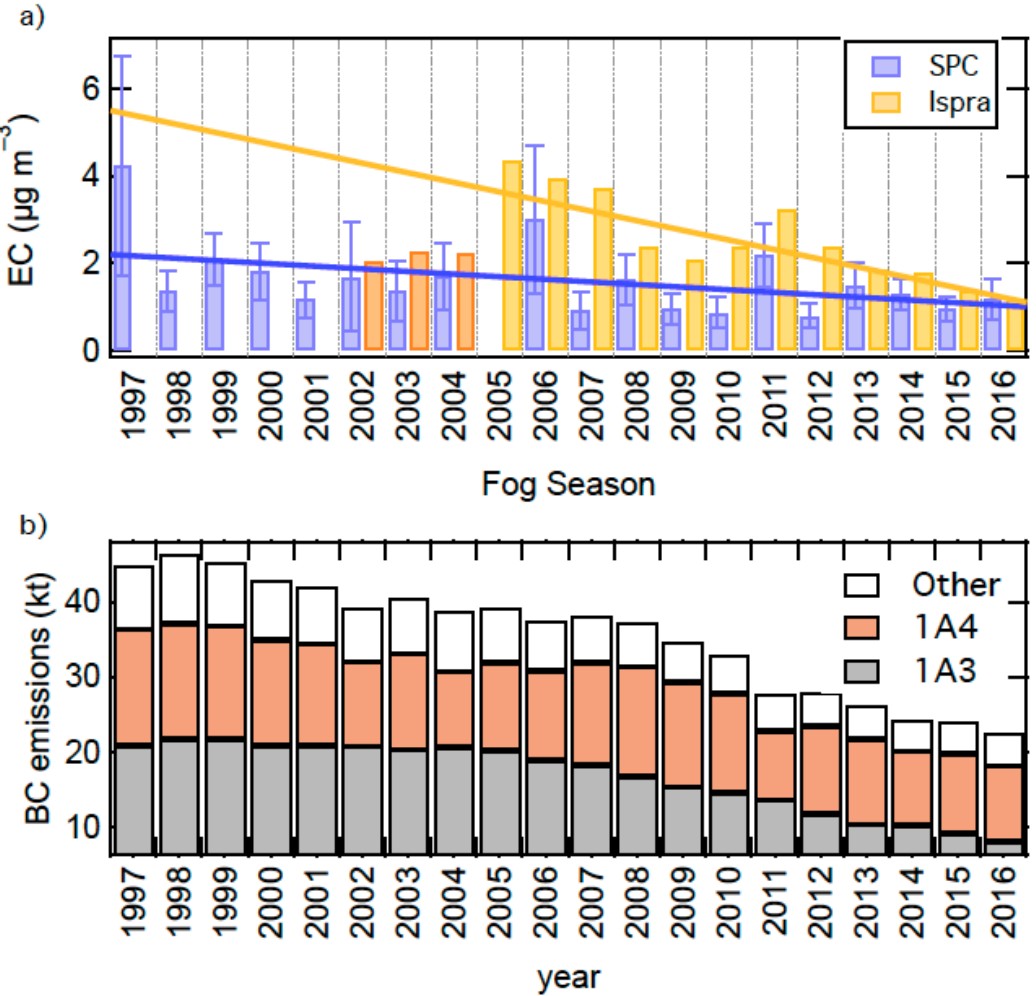

**Figure 5.** (**a**) Trend of atmospheric EC average concentration estimated from fog water composition in SPC (blue bars) and from direct EC measurements in Ispra from November to March of the following year (dark orange bars for 2002–2004 seasons, and light orange bars for 2005 and following seasons); (**b**) trend of national black carbon emissions from 1997 to 2016, compiled by the Institute of Environmental Protection and Research (ISPRA); contributions from non-industrial combustion (1A4 in brown) and road transport (1A3 in grey) are highlighted.

**Table 2.** Statistical parameters of Kendall rank correlation and Theil-Sen analysis for the EC time series at San Pietro Capofiume (SPC 1997–2016) and Ispra (2003–2016).

|       | Kendall's $\tau$ | Sen's Slope | Mean Absolute Deviation (MAD) |
|-------|------------------|-------------|-------------------------------|
| SPC   | −0.43            | −0.06 $\mu g\ m^{-3}\ y^{-1}$ | 0.03 $\mu g\ m^{-3}\ y^{-1}$ |
| Ispra | −0.69            | −0.22 $\mu g\ m^{-3}\ y^{-1}$ | 0.09 $\mu g\ m^{-3}\ y^{-1}$ |

Consistent long-term records of atmospheric EC concentration are quite limited worldwide. Murphy et al. [32] investigated the trends of EC at rural and remote sites across the United States from 1990 to 2004 and reported, for low-altitude sites, a decreasing trend ranging between zero and 6% per year. The trend was more prominent during the cold season. Higher reduction rates are reported for urban locations. Yamagami et al. [33] observed a decreasing trend of 13% per year over the period 2003–2016 in the urban area of Nagoya (Japan). Shorter time series, although less representative, still confirm a decreasing tendency over the more recent years. For example, atmospheric EC concentration decreased on average by 10%–20% from 2012 to 2017 in the Los Angeles area, with the highest reduction during the cold season (up to 28%) [34]. Singh et al. [35] reported a statistically significant reduction in

EC at the urban and curbside stations across the United Kingdom, ranging between 6% and 8% per year over the period 2009 to 2016. Although representative exclusively of the cold season (when fog is present), in the present study, we estimate a decrease in atmospheric EC of 4% per year, on average, in agreement with observations at US rural sites [32]. Such a rate is also in agreement with the reduction in BC emissions at national level, reported in Figure 5b [36]. Emissions are calculated at the national level every year by the Italian Institute for Environmental Protection and Research (ISPRA). According to the most recent inventory, emission data are characterized by a clear decreasing trend over the period 1997–2016, with steeper reductions after 2008. Sectors for which really high decreases are observed are: (i) combustion in energy and transformation industries; (ii) production processes and road transport; and (iii) other mobile sources; while an increase was observed from the non-industrial combustion plants. Long-term atmospheric EC data confirm the effectiveness of some air quality policy measures implemented during the last 20 years, especially in reducing emissions from road transport [37]. Additional measures will be implemented in this direction, according to the results of air emission and air quality scenarios.

## 4. Conclusions

The measurement of the EC concentrations in fog water samples collected over a period of ca. 20 years at the station of San Pietro Capofiume allowed the reconstruction of a long-term trend of aerosol EC in the southeastern part of the Po Valley, Italy.

Over the study period, a reduction in EC aerosol concentration of about 4% per year was estimated, indicating the effectiveness of the implemented policy aimed at emission reduction. Similar trends are reported for PM; in fact, Bigi et al. [38] calculated that $PM_{10}$ in the Po Valley decreased by 2%–5% per year over the period 2002–2011. $PM_{2.5}$ atmospheric concentrations have been routinely monitored only more recently [39]. Trend analysis across the Po Valley identifies a decreasing tendency between 1% and 8% per year since the mid-2000s [15,39]. Longer time series analysis of total suspended particles (TSP) reported even a stronger decreasing trend from the end of the 1990s.

The concentration of aerosol EC at the rural station of San Pietro Capofiume is comparable to that of urban sites across Europe and of Eastern Europe locations. This is also true for the Ispra station that is located in the northern part of the Po Valley. These results confirm that the Po Valley is clearly a pollution hotspot, not only in its highly urbanized environment, but also in rural locations. This results from a combination of the effects of the diffuse sources of pollution, and also by the Po Valley orography that prevents pollution dilution.

**Author Contributions:** Conceptualization, S.G. and S.F.; methodology, S.G. and S.S.; data curation, S.G. and L.T.; formal analysis, S.G.; investigation, L.T., S.S., P.I., D.C., J.-P.P., F.C., V.P., D.B., C.L., A.G. and L.L.; writing—original draft preparation, S.G. and S.F.; writing—review and editing, S.G. and S.F. All authors have read and agreed to the published version of the manuscript.

**Funding:** This research received no external funding.

**Acknowledgments:** The authors would like to acknowledge Riccardo De Lauretis (ISPRA) for providing the black carbon national emission inventory, and Fabio Savelli (CNR-ISMAR) for total carbon analyses.

**Conflicts of Interest:** The authors declare that there are no conflict of interest.

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
