# Peer review of "Reconstructing Elemental Carbon Long-Term Trend in the Po Valley (Italy) from Fog Water Samples"

_atmosphere, doi:10.3390/atmos11060580_

Round 1
Reviewer 1 Report
Fog water chemical measurements was used to reconstruct possible Elemental Carbon (EC) trend in the Po Valley from 1997 to 2016. Based on the measured total carbon (TC) and EC in fog water from two different periods, an average EC to TC ratio for the EC measurement gaps between 2000 and 2012. For simplifying the process over a long period, this study assumes that the EC is completely scavenged by fog droplets. Therefore, the atmospheric loading of EC by fog droplets could be the product of the concentration of EC suspended in fog water and the atmospheric liquid water content. The reconstructed results show a decreasing trend over the past two decades. The authors conclude that the rural areas in Po Valley has higher concentration of EC compared with other European rural sites.
Comments and Suggestions
- Are the two measurement sites (i.e., San Pietro Capofiume (SPC) and Ispra) considered as representative sites for the Po Valley? Please describe more relevant studies for both sites and include major seasonal wind directions for the Po Valley, including both measurement sites if possible.
- In the conclusion, the EC concentrations in SPC (a rural sites in the Po Valley) are observed higher compared with European rural sites. Any possible reasons or potential sources?
- In the Abstract (line 31), please verify the value of decreasing trend. Is it 6% per year? Which is different from the conclusion (see line 323) that about 4% per year was estimated.
- In the Table 1 (line 297 to 298), please include the units for Kendall’s tau, Sen’s slope and Mean Absolute Deviation.
- Please verify the format of the reference 18 (see line 389).
Author Response
Reviewer comment in bold
Are the two measurement sites (i.e., San Pietro Capofiume (SPC) and Ispra) considered as representative sites for the Po Valley? Please describe more relevant studies for both sites and include major seasonal wind directions for the Po Valley, including both measurement sites if possible.
We added a paragraph to section experiments entitled “Measurements sites”. The paragraph includes the description of the two sites, SPC and Ispra, and the typical meteorology during the winter seasons. Additional references are added to better describe the site representativeness. The number of the paragraphs in section “Experiments” is modified accordingly.
In the conclusion, the EC concentrations in SPC (a rural sites in the Po Valley) are observed higher compared with European rural sites. Any possible reasons or potential sources?
The high pollutant concentration in the rural areas of the Po valley, compared to other European rural sites, is due to multiple concomitant factors that define the Po valley as a highly polluted region (EEA 2014). In particular, the Po valley is densely populated and most of the industrial and agricultural activities of the country are located in this area. Finally, the orography of the territory makes difficult the pollution dispersions, especially during fall and winter. We acknowledge that some relevant details about the Po valley were missing in the first submission.
In the Abstract (line 31), please verify the value of decreasing trend. Is it 6% per year? Which is different from the conclusion (see line 323) that about 4% per year was estimated
The abstract was corrected. The exact decreasing trend is 4% on average per year.
In the Table 1 (line 297 to 298), please include the units for Kendall’s tau, Sen’s slope and Mean Absolute Deviation
The unit of Sen’s slope and MAD were added to the table.
Please verify the format of the reference 18 (see line 389)
The format was corrected
Reviewer 2 Report
Please see the attached document for comments.

Author Response
Reviewer comments in bold
First, I find the conversion to atmospheric concentrations a bit questionable and unnecessary. While the approach to determining a factor to convert fog water EC to atmospheric concentrations using measurements of ambient EC is appropriate, the regression presented in Figure 5 suggest there is not a strong enough relationship between these two datasets to yield a reliable factor. Further, the fact that the linear fit to these data (the slope of which is taken to be the rescaling factor) only intersects one of the data points is concerning. This brings the validity of the estimated atmospheric values into question, even if they fall within a range that is consistent with the literature. I recommend using the fog EC concentrations instead to show the trend in EC. These values should still show a 4% decline which is consistent with values reported in the literature and trends in BC emissions. Comparisons between the fog concentrations reported here and those reported for fog or precipitation in the literature should also be included
We believe that the conversion of fog water concentration into atmospheric concentration is one of the key points of this study, since it aims at reconstructing long term trend of atmospheric concentrations, to discuss air quality policy improvements, and potentially in future work, climate implications. Nevertheless, we agree with the referee that the comparison of fog water concentration with atmospheric concentration for each single fog event is not useful and can be misleading for this study purpose. In fact, this comparison is affected by scavenging efficiency of elemental carbon particles, which depends on particles mixing state, and vary from event to event (Gilardoni et al., 2014). In addition, it is reasonable to assumes that the ratio between fog water and atmospheric concentration depends on the duration of each single fog events, which might range from a few hours to more than one day. These factors together explain the large scatter observed in figure 5. Since the purpose of this work is to derive seasonal average atmospheric concentration from fog water composition, we estimate the conversion as the average of seasonal fog water to atmospheric EC concentration ratios. We believe that averaging over a whole season takes into account the event inter-variability, and better support the purpose of investigating the long-term tend. The text was modified, Figure 5 was removed and Figure 6 was re-numbered accordingly.
I do believe a comparison with the ambient EC measurements is a good addition to the paper but should be done so in a different way. I suggest instead including the seasonally-averaged ambient values on Figure 6 to demonstrate how trends in these data compare to both trends in fog EC and ambient EC at Ispra.
We decided to keep the conversion of fog water concentration into atmospheric concentration, so a direct comparison is presented in figure 6 (now re-numbered as figure 5).
While I think using a mean EC/TC ratio to determine the EC concentrations in fog samples collected between 2001 and 2012 is appropriate and necessary, it is not clear if the variability in the EC/TC ratio is considered in the analysis. If this has been done, it should be stated in the text. If not, some effort (e.g., propagation of error) should be made to account for the variability in EC/TC in the 2001-2012 EC estimates. It is also interesting to note that the variability in EC/TC is larger between 2012-2016 compare to 1997-2001. Can the authors comment on why this is
We agree with the referee about the opportunity to explicitly indicate the uncertainty introduced by the assumption of a constant EC to TC ratio and a constant conversion factor to derive atmospheric concentration from fog water composition. The uncertainty associated to the EC to TC ratio is added to the text and the uncertainty of the seasonal average atmospheric EC is estimated by error propagation. The t-test analysis indicates that the two datasets have similar distributions.
Are multiple samples taken per fog event?
Each fog event corresponds to a single fog water sample. This detail is now added to the text, in section “Fog water collection”
The Ispra site sites rather far away from the SPC site. Given the likely significant differences in proximity to emissions (which the authors note) and meteorology, it is unclear why seasonal EC values are compared between the sites. Some text should be added to explain the reason for using data exclusively from Ispra and not other sites in the Po Valley.
The Ispra site is one of the few sites in Europe where EC measurements have been performed routinely for over a decade. In addition, Ispra is a regional site, so we expect it to be representative, not just of the nearest surrounding area. To clarify the significance of the comparison with Ispra, a new paragraph was added.
In the discussion of the Cavalli et al. and Zanatta et al. results (lines 256-260), are the authors saying that EC at SPC in wintertime was 0.1-0.6? If yes, some text is needed to explain the discrepancy between these values and those determined by this analysis. If not, this statement needs reworking.
We agree with the referee on the need to reword the sentence and modified the text. In addition, to improve the clarity, we added a table reporting the EC concentration values discussed in the text.
The Theil- g m-3 decade -1 in the text (lines 292-293) but 0.06 in Table 1. Which values is correct?
Both values are correct since we observed a decrease of 0.06 mg m-3 per year, which corresponds to 0.6 mg m-3 per decade, but than the MAD indicated is not be correct. To avid confusion, the sentence was corrected.
Similarly, a reduction in EC of 6% per year is reported in the abstract but 4% is mentioned in the Results/Discussion (lines 310-312). Which value is correct?
The value erroneously reported in the in the abstract was corrected to 4%
There are also several typos in the text that need to be addressed (e.g., line 40 thei should be their).
Manuscript has been carefully reviewed for typos.
Reviewer 3 Report
The publication is very valuable research material although its character more indicates its reporting nature. The authors reliably developed the results of the research, but in my opinion, however, there is no explanation for the occurring phenomena. There is no interpretation of the results supplemented with meteorological data and on sources of emissions. I therefore believe that the publication needs to be supplemented with the above mentioned aspects and analyzes.
Detailed comments were presented in the commentary mode.

Author Response
Reviewer comments in bold
Line 156: Season averaged TC - from what number of n-samples was this value obtained?
The number of fog samples collected during each season varied between 7 and 32, with an average of 20 samples for season. To add these details, the sentence at the end of section “Fog water collection” was modified. In addition, the number of samples that refer to the average values reported in line 156 are now specified in brackets.
Line 262: And whether specific impact of phenomena of a meteorological nature and topography of the area? Trajectory analyzes, e.g. using the Hysplit model for single episodes, could also complement the interpretation.
We thank the reviewer for pointing out the need of a better clarification of the peculiar topography and meteorological conditions that characterise the Po valley. The higher concentration of EC observed in the Po valley are partially attributed to the orography of the valley, which prevent pollutant dilution, and meteorology, which favour pollutant accumulation, especially during winter. In order to emphasize these effects, a sentence was added to the section “Sampling sites”. Also, in a previous work, high spatial resolution back-trajectories (0.2 x0.2°) were analysed with Potential Source Contribution Function tool (Gilardoni et al., 2011). The analysis identified the northern Po valley as the main area impacting carbonaceous aerosol concentration in Ispra. The sentence in line 261 was therefore modified.
Line 265: In my opinion, a table comparing and the results of data from other European countries would be valuable.
We agree with the reviewer that a table comparing the result of this study with previous works would be valuable and this table was added to section “Long term trend of atmospheric EC concentrations”
Figure 6: The observed trend is not of a linear nature - the polynomial function seems to be optimal.
The line reported in figure 6 is the results of the Theil-Sen slope estimation, which implicitly assumes a linear trend. We chose the Theil-Sen estimator since the purpose of this work is to detect a potential average decreasing trend and its slope, which based on emission estimation, we expect to be well approximated by a linear regression. The non-linearity of EC observations in Ispra could be affected by the different experimental method adopted for the EC quantification from 2002 to 2004, so we did not perform the fitting with a polynomial function, to avoid the overweighting of the different analytical method
Line 315: In my opinion, a citation in the form of a number should be used.
The reference was added to the reference list.
Line 318: Please elaborate on this thought and support concrete research results in connection with transport emissions data.
We agree that this last paragraph needs to be better supported and the paragraph was therefore modified.
Round 2
Reviewer 2 Report
The authors have done an excellent job addressing my concerns in their revised manuscript. I believe the manuscript is nearly ready for publication but could use some additional very minor editing for grammar and spelling. I've listed a few suggestions below.
-Line 21-22: commas needed after “(EC)” and after “combustion”
-Line 43: “as” needed before “an unavoidable…”
-Line 76: “next” should be “near future…”
-Section 2.1: "valley" should be capitalized.
-Line 144: “for season” should be “per season”.
-Lines 287-288: “Two show...” should be “to show” and “tow series” should be “two”
-Line 288-290: Need “The” before "Ispra monitoring site...". The sentence in general could use some reworking.